# Novel Visual Category Discovery with Dual Ranking Statistics and Mutual Knowledge Distillation

**Bingchen Zhao**[1]    **Kai Han**[2,3,4*]

[1]Tongji University  [2]The University of Hong Kong  [3]Google Research  [4]University of Bristol
zhaobc.gm@gmail.com   kaihanx@hku.hk

## Abstract

In this paper, we tackle the problem of novel visual category discovery, *i.e.*, grouping unlabelled images from new classes into different semantic partitions by leveraging a labelled dataset that contains images from other different but relevant categories. This is a more realistic and challenging setting than conventional semi-supervised learning. We propose a two-branch learning framework for this problem, with one branch focusing on local part-level information and the other branch focusing on overall characteristics. To transfer knowledge from labelled data to unlabelled data, we propose using dual ranking statistics on both branches to generate pseudo labels for training on the unlabelled data. We further introduce a mutual knowledge distillation method to allow information exchange and encourage agreement between the two branches for discovering new categories, allowing our model to enjoy the benefits of global and local features. We comprehensively evaluate our method on public benchmarks for generic object classification, as well as the more challenging benchmarks for fine-grained visual recognition, achieving state-of-the-art performance.

## 1   Introduction

Superior performance on many problems has been achieved by recent machine learning models, especially the ones based on deep learning. While the success comes at the cost of large-scale human annotation, which is prohibitively expensive in practice. For example, modern convolutional neural networks (CNNs) can surpass human-level recognition performance on ImageNet after training with over one million labelled images [22]. On the one hand, it is not possible to annotate all possible classes in the real world, as there are way more classes than the $1,000$ classes in ImageNet and new classes keep growing over time. On the other hand, annotating specific data, such as the medical data, may require specific expertise, which renders the large-scale annotation extremely difficult, if not impossible. Therefore, it is desired to enable the machine learning systems to deal with unlabelled data automatically.

Recently, the problem of novel category discovery was formalized in [20, 18], which aims at discovering new visual categories on unlabelled data by transferring knowledge from labelled data. The labelled data is assumed to contain similar but different categories to those in the unlabelled data. This problem is similar to semi-supervised learning in the sense that both labelled and unlabelled data are used to learn the model. While novel category discovery is much harder because semi-supervised learning assumes that every class contains labelled instances, while for novel category discovery there are no labels available for the new classes in the unlabelled data. This setting is also relevant to

---

* Corresponding author.

35th Conference on Neural Information Processing Systems (NeurIPS 2021)

unsupervised clustering. But differently, novel category discovery makes use of the labelled data to extract a specific class prior (*i.e.*, the properties that delineate a class) for partitioning the unlabelled data, while the unsupervised clustering may produce multiple different but equally valid clustering results by adopting different properties (*e.g.*, color, shape, pose, lighting, etc). In this paper, we introduce a simple and effective two-branch framework for novel category discovery, with one branch focusing on local part-level information and the other branch focusing on overall characteristics. Our contributions are as follows.

First, we propose to apply dual ranking statistics for transferring knowledge from the known classes in the labelled data to the unlabelled data, resulting in more robust pseudo label generation for learning on the unlabelled data. We maintain a dynamic object part dictionary and apply part-level ranking statistics on the similarity distribution of each instance over the dictionary to obtain pseudo labels, which are complementary to the pseudo labels obtained by simply examining the ranking statistics of global descriptors.

Second, we introduce a mutual knowledge distillation method to allow information exchange and encourage agreement between the local and global branches. The dual ranking statistics provide global and part-level information for learning in two branches separately. The mutual knowledge distillation further allows information exchange between the two branches and makes them benefit from each other. Unlike conventional methods that distill between teacher and student models with known labels, our method distills between two branches of the same model without any manual annotations.

Third, we comprehensively evaluate our method on public benchmarks for generic object classification, including CIFAR10, CIFAR100, and ImageNet, obtaining state-of-the-art results. Furthermore, we also validate our approach on the more challenging fine-grained datasets CUB-200, Stanford-Cars, and FGVC-Aircraft, in which the local details are more important to distinguish different classes. Our method outperforms existing methods by a substantial margin, thanks to the ability of our model to subtly exploit both local and global information. Our code can be found at `https://github.com/DTennant/dual-rank-ncd`.

## 2 Related work

Our work is relevant to novel category discovery, knowledge distillation, and part-level feature learning. We briefly review the most relevant work below.

*Novel category discovery* is a relatively new problem setting recently formalized by [20, 18] with the task being automatically discovering new object categories in the unlabelled data by making use of a labeled dataset containing different but relevant object classes. The purpose of using the extra labelled data is to learn a category prior to reduce the ambiguity of class definition. This task is closely related to unsupervised clustering and semi-supervised learning, but also significantly different from them. Unsupervised clustering has been studied for decades with many classical approaches [35, 3, 9] and deep learning based solutions [50, 16, 40] being proposed. Due to the lack of a proper class prior, multiple equally valid clustering results can be achieved by different criteria. Therefore, it can not be directly applied to discovery new classes as we expect the model to follow a unique class definition. In semi-supervised learning [43, 39, 6, 38, 46], unlabelled data are used together with labelled data to train a more robust model, with the assumption that all classes in the unlabelled data have labelled instances. However, this is unlikely to be true for real applications where unlabelled data may come from new classes. Thus, novel category discovery is a more realistic setting. The DTC method introduced in [20] approaches this problem in two steps. The model is firstly trained with supervised learning on the labelled data to capture high-level semantic class information and then trained on the unlabelled data with a clustering loss to discover new categories. In [18, 19], Han *et al.* proposed to use self-supervision to bootstrap feature learning and introduced ranking statistics on the feature embedding to provide pseudo labels for training on unlabelled data. The KCL [25] and MCL [26] methods were designed for general transfer learning across domains and tasks, which can also be applied for novel category discovery. These two methods maintain two models for training and testing, a pretrained binary classifier for pseudo label generation and a clustering model. The binary classifier pretrained on the labelled data is used to provide pseudo labels to train the clustering model on unlabelled data. Concurrent to our work, several methods [28, 58, 57, 14] are proposed to improve the performance of novel category discovery from different perspectives, showing promising results.

The existing methods only consider the global feature descriptors while ignoring the local object parts which are essential to distinguish classes that look similar. In our approach, we jointly consider both and allow information exchange between them for more reliable new category discovery.

*Knowledge distillation* is often employed to learn a compact student model using the knowledge distilled from a larger teacher model by enforcing the agreement of outputs or representations between the two models (*e.g.*, [24, 41, 54, 52, 27, 29, 1, 48]). Apart from distilling a larger teacher model to a smaller student model, self-distillation methods show that distilling between two identical models can also improve the performance [53, 15, 4]. Mutual learning [56] is a similar technique that trains two models of the same architecture simultaneously but with different initialization and encourages them to learn collaboratively from each other. It has been shown that different initialization leads the model to focus on different regions of the input and thus the trained model can better capture the holistic structure of the data, resulting in better performance. Though effective, the above methods require the labels for training. Thus they can not be applied for novel category discovery. Recently, [13] introduced a self-supervised distillation method called SEED to improve the representation learning on a small model (student) by distilling from a pretrained large model (teacher). The student model is trained to predict the same similarity score distribution inferred by the frozen teacher model over a queue of instances. We draw inspiration from [13] to design the mutual knowledge distillation module in our method for novel category discovery between the local and global branches.

*Part-level features* have been shown to be effective for tasks involving image verification such as few-shot learning [10, 55, 12] and image retrieval [7, 34, 5, 2, 37, 42, 45]. In [10], the part-level visual concepts are generated by clustering feature vectors from the feature maps obtained using a trained neural network. These visual concepts are used as matching primitives at test time for few-shot learning. Zhang *et al.* [55] used the Earth-Mover's Distance (EMD) between local part features to measure the distance between two images for matching. Doersch *et al.* [12] proposed CrossTransformers to model the cross-attention between each local part of a query image and a set of support images for few-shot learning. Part-level features have been more widely used in the domain of image retrieval with both classical [34, 5] and deep learning based [2, 37, 42, 45, 7] methods. Intuitively, the part-level features provide better precision for the retrieved results because the local matching is more restricted, and global features yield better recall [7]. In our work, we leverage both local and global features to enjoy the benefits of both through the dual ranking statistics and the mutual knowledge distillation, for more robust novel category discovery.

## 3 Method

The goal of novel category discovery is to automatically partition unlabelled instances $x_i^u \in \mathcal{D}^u$ into $C^u$ semantic clusters by transferring the knowledge from the labelled instances $(x_i^l, y_i^l) \in \mathcal{D}^l$, where $y_i^l \in \{1, \ldots, C^l\}$. The key assumption is that the classes in $\mathcal{D}^l$ and $\mathcal{D}^u$ are relevant though different, so that the properties that constitute a class learned from $\mathcal{D}^l$ can be transferred to $\mathcal{D}^u$. Following [18], we assume $C^u$ is known a priori. When it is unknown, we can employ off-the-shelf methods such as [20] to get an estimate.

To tackle this challenge, we introduce a framework with mutual knowledge distillation between a global branch and a local branch (see fig. 1). The global branch is designed to capture the overall feature, while the local branch is designed to focus on each individual spatial local part. They have a shared feature extractor $f_\theta$. Each branch has a feature projection layer and two linear heads. We denote the feature projection layer as $\psi$ and the two linear heads as $\eta^l$ and $\eta^u$ respectively. We also denote $\psi$ followed by the average pooling operation as $\bar{\psi}$. Unless stated otherwise, we use subscripts $g$ and $p$ to differentiate feature projection layers and linear heads of global and local branches respectively in the rest of the paper. The two linear heads are responsible for classifying $C^l$ labelled categories and clustering $C^u$ unlabelled categories respectively. To transfer knowledge from the labelled data to the unlabelled data, we adopt global image level and local object part-level ranking statistics on the feature representations. In this way, the model will have reliable pseudo labels for training on the unlabelled data. Moreover, we also incorporate mutual knowledge distillation between the two branches to enforce global and local agreement, leading to more reliable novel category discovery. Next, we will introduce each component of our framework in more detail. In section 3.1, we first introduce the our knowledge transfer method using dual ranking statistics. In section 3.2, we describe our mutual knowledge distillation method for novel category discovery. Lastly, we summarize the overall training loss in section 3.3.

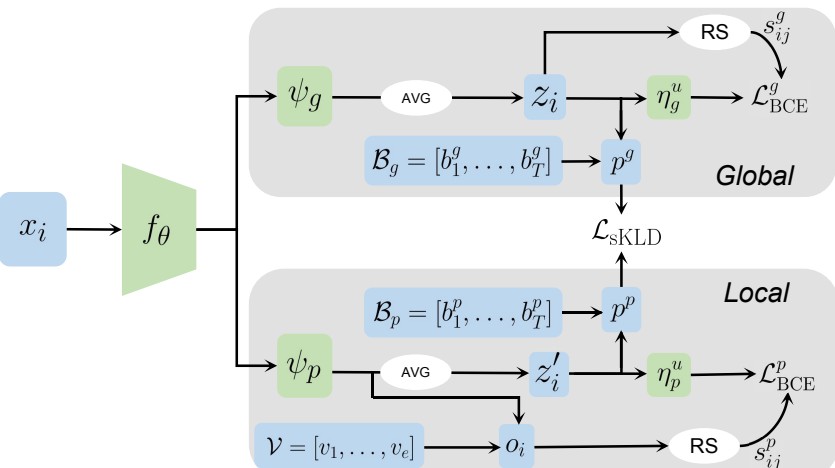

Figure 1: **Overview of our proposed method.** An input image $x_i$ is firstly processed by the feature extractor $f_\theta$. The resulting feature is sent to the global and local branches. The global branch uses ranking statistics on the global feature embedding $z_i$ to generate pseudo labels to the $\mathcal{L}^g_{\text{BCE}}$ loss for new class discovery, while the local branch uses ranking statistics on the similarity score vector $o_i$ between each of the local parts and a dynamic part dictionary to generate pseudo-labels. The two branches mutually distill knowledge from each other to allow information exchange and encourage agreement through two auxiliary memory banks with $\mathcal{L}_{\text{sKLD}}$ loss. We omit the supervised linear head, cross-entropy loss $\mathcal{L}_{\text{CE}}$ and consistency loss $\mathcal{L}_{\text{MSE}}$ in the plot for the sake of clarity.

## 3.1 Dual ranking statistics for knowledge transfer

It is proven that ranking statistics is robust to noise, especially in high-dimensional space [51]. Han *et al.* [18] proposed to use ranking statistics for novel category discovery. The idea is to generate pair-wise pseudo labels by comparing the top-$k$ ranks of two feature vectors via examining the feature magnitude. Specifically, the binary pseudo label $s_{ij}$ is determined by

$$s_{ij} = \mathbb{1}\left\{\text{top}_k(z_i) = \text{top}_k(z_j)\right\}, \tag{1}$$

where $z_i$ and $z_j$ are feature vectors of two unlabelled images. With the binary pseudo labels, the model can then be trained using the binary cross-entropy loss on the unlabelled data.

Unlike [18] which only considers global image feature descriptors, we also explore each individual object part for novel category discovery. Verification based on part-level features has been shown to be effective to match two images (*e.g.* [55, 12]). The pair-wise verification using global features focuses on the holistic structure of the object and thus may introduce more false positives with high recall (but low precision). While pair-wise verification using local part features focuses on local details, which is more strict, and thus may introduce more false negatives with high precision (but low recall) [7]. Hence, local part features and global features are complementary to each other for verification and should be considered jointly for more robust category discovery, as ideally we expect to have both high precision and recall. Therefore, to achieve this goal, we propose to have one branch using global features and another branch using part-level features. Ranking statistics is applied on each branch to generate pseudo labels. For the global one, we simply apply a soft extension of the hard ranking statistics [18]. In particular, instead of forcing $s_{ij}$ in eq. (1) to be either 0 or 1, it can be replaced by a continuous value $s_{ij} = \frac{c}{k} \in [0, 1]$ where $c$ is the number of shared elements in $\text{top}_k(z_i)$ and $\text{top}_k(z_j)$. Hence, the BCE loss for the global branch can be written as

$$\mathcal{L}^g_{\text{BCE}} = -\frac{1}{M^2}\sum_{i=1}^{M}\sum_{j=1}^{M}[s^g_{ij}\log\eta^u_g(z^u_i)^\top\eta^u_g(z^u_j) + (1-s^g_{ij})\log(1-\eta^u_g(z^u_i)^\top\eta^u_g(z^u_j))], \tag{2}$$

where $M$ is the number of unlabelled images, $z^u_i = \bar{\psi}_g(f_\theta(x^u_i)) \in \mathbb{R}^d$ is a global feature vector of the image $x^u_i$, and $s^g_{ij}$ is the soft ranking statistics score between $x^u_i$ and $x^u_j$.

For the local one, we propose to maintain a memory bank to act as a dynamic object part dictionary and obtain the ranking statistics by comparing each object part with the collection of part descriptors in the memory bank. Concretely, the memory bank is a First-In-First-Out (FIFO) queue $\mathcal{V} = [v_1, \ldots, v_e]$

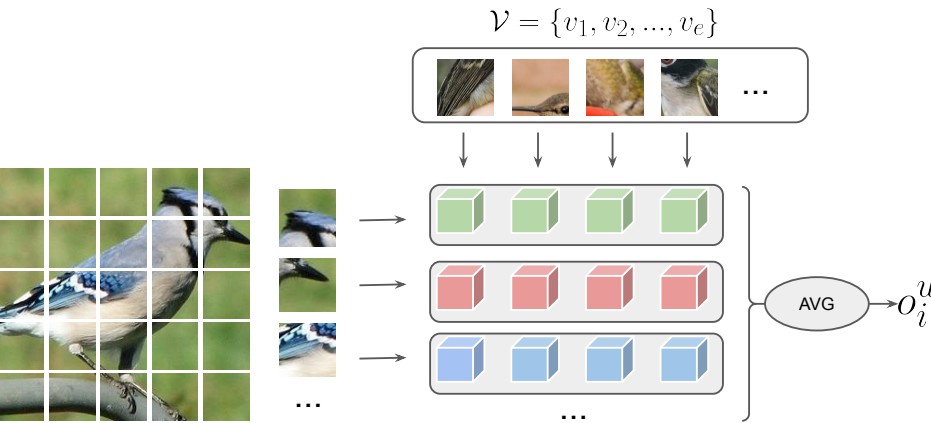

$$\mathcal{V} = \{v_1, v_2, ..., v_e\}$$

Figure 2: **Design of the local comparison process**. Each part of an image is compared against all parts in the memory bank $\mathcal{V}$, forming one similarity vector for each part. All resulting similarity vectors are then fused to a single vector by average pooling, which is used to generate pair-wise pseudo labels.

storing the part-level features, where $e$ is the number of part-level features in the queue. Each part feature in $v$ is the $d$-dimensional feature vector from a randomly selected spatial location in the feature map $\psi_p(f_\theta(x_i)) \in \mathbb{R}^{d \times h \times w}$ of an randomly sampled image $x_i \in \mathcal{D}^l \cup \mathcal{D}^u$. We select one part feature vector from each image in current mini-batch. Other part sampling methods like using all parts or selecting based on activation maps can also be applied (as will be seen in section 4.3), while we found that the simple random sampling is on par with other more complicated methods. Following MoCo [21], when the current mini-batch of part features is enqueued into $\mathcal{V}$, the oldest mini-batch in $\mathcal{V}$ is dequeued. For the feature vector $q_j^u$ drawn at every spatial location of the feature map $\psi_p(f_\theta(x_i^u))$ in current mini-batch, we can then obtain a $e$-dimensional vector representing the similarity between $q_j$ and all part features in the dictionary $\mathcal{V}$. All $q_j$ for $x_i^u$ across $h \times w$ locations are then fused together by average-pooling. Let us denote by $o_i^u \in \mathbb{R}^e$ the fused similarity vector (see fig. 2). For any pair of unlabelled images $x_i^u$ and $x_j^u$ in current mini-batch, their soft ranking statistics score $s_{ij}^p$ can then be obtained by comparing $o_i^u$ and $o_j^u$ as described above for the global feature descriptors. Similar to eq. (2), we can define the BCE loss $\mathcal{L}_{\text{BCE}}^p$ to train the local branch. The BCE loss for both branches can then be written as

$$\mathcal{L}_{\text{BCE}} = \mathcal{L}_{\text{BCE}}^g + \mathcal{L}_{\text{BCE}}^p. \tag{3}$$

## 3.2 Mutual knowledge distillation for novel category discovery

So far, our model considers local and global information in two branches independently. To make the two branches directly benefit from each other, we propose a mutual knowledge distillation method for novel category discovery to allow information exchange and encourage agreement between the local and global branches.

Existing mutual knowledge distillation methods are normally designed for two models (*i.e.*, teacher model and student model or two peer models) [24, 56], while we distill between two branches of the same model with each branch having a different focus. More importantly, unlike the conventional mutual learning and knowledge distillation methods that have the same class assignment for the same input due to the availability of labels, in our setting, the cluster assignments of the unlabelled linear heads $\eta_g^u$ and $\eta_p^u$ for the same unlabelled data point may be completely different for the two branches. Therefore, these knowledge distillation methods can not be applied for novel category discovery. Recently, [13] introduced the knowledge distillation between the teacher and student models by comparing the similarity score distribution between each instance and a queue of features from the larger teacher model. Inspired by [13], we develop a mutual knowledge distillation method between the local and global branches for novel category discovery without using labels.

Instead of using the softmax output of $\eta_g^u$ and $\eta_p^u$ for mutual learning, which is invalid in our case, we mutually distill the two branches via the similarity score distribution over a memory bank per branch. Specifically, we maintain two FIFO feature banks to store the features extracted by $\bar{\psi}_p$ and $\bar{\psi}_g$, denoted as $\mathcal{B}_p = [b_1^p, \ldots, b_T^p]$ and $\mathcal{B}_g = [b_1^g, \ldots, b_T^g]$. Both $\mathcal{B}_p$ and $\mathcal{B}_g$ contain features

of $x_i \in \mathcal{D}^l \cup \mathcal{D}^u$. Again, similar to MoCo [21], when the current mini-batch is enqueued, the oldest mini-batch is dequeued in both memory banks. For each unlabelled image $x_i^u$, we first obtain the feature representations using $z_i^u = \bar{\psi}_g(f_\theta(x_i^u))$ and $z_i'^u = \bar{\psi}_p(f_\theta(x_i^u))$. The similarity score distribution $p^g(x_i^u, \mathcal{B}_g)$ and $p^p(x_i^u, \mathcal{B}_p)$ can be defined as

$$p^g(x_i^u, \mathcal{B}_g) = [p_1^g, \ldots, p_T^g], \quad p_j^g = \frac{exp(z_i^u \cdot b_j^g/\tau)}{\sum_{b_k^g \sim \mathcal{B}_g} exp(z_i^u \cdot b_k^g/\tau)} \tag{4}$$

and

$$p^p(x_i^u, \mathcal{B}_p) = [p_1^p, \ldots, p_T^p], \quad p_j^p = \frac{exp(z_i'^u \cdot b_j^p/\tau)}{\sum_{b_k^p \sim \mathcal{B}_p} exp(z_i'^u \cdot b_k^p/\tau)} \tag{5}$$

where $\tau$ is a scalar temperature to control the sharpness of the similarity score distribution.

To encourage agreement between the similarity score distributions of an unlabelled image from two branches, we apply mutual learning on the score distribution using the symmetric Kullback-Leibler Divergence (sKLD) loss

$$\mathcal{L}_{\text{sKLD}} = \frac{1}{2}(D_{KL}(p^p \| p^g) + D_{KL}(p^g \| p^p)) \tag{6}$$

where $D_{KL}$ is the Kullback–Leibler (KL) divergence with $D_{KL}(p_1 \| p_2) = p_1 \log \frac{p_1}{p_2}$. In this way, the global and local branches of the model can learn from each other, while maintaining their own merits. Our mutual knowledge distillation method differs from SEED [13] in several aspects. First, SEED aims at improving high level visual representation, while we aim at enforcing the agreement between local and global features for novel category discovery; second, SEED distills between two different models while we distill between two branches of the same model; third, SEED maintains a queue derived from the global features of the teacher model, while we maintain two queues as dictionaries for global and local features; last, in SEED the larger teacher model is frozen during distillation and the student model is trained with cross-entropy loss, whereas in our method both branches are jointly trained with the sKLD loss.

### 3.3 Overall training loss

Apart from the BCE and sKLD losses introduced above, we also apply the standard cross-entropy loss on both branches, which is written as

$$\mathcal{L}_{\text{CE}} = -\frac{1}{N} \sum_{i=1}^N y_i \log \eta_g^l(z_i^l) + y_i \log \eta_p^l(z_i'^l) \tag{7}$$

where $N$ is the number of labeled images, $z_i^l = \bar{\psi}_g(f_\theta(x_i^l))$, and $z_i'^l = \bar{\psi}_p(f_\theta(x_i^l))$. Similar to [18], the feature extractor $f_\theta$ is pretrained with self-supervised learning and is frozen during the training for novel category discovery.

Following [18, 20], we also include the consistency regularization loss to enforce the predictions of the same data point under different transformations to be the same. Specifically, let $\hat{x}_i$ be a randomly transformed counterpart of the input image $x_i$. The consistency loss is then defined as

$$\mathcal{L}_{\text{MSE}} = \frac{1}{N} \sum_{i=1}^N [(\eta_g^l(z_i^l) - \eta_g^l(\hat{z}_i^l))^2 + (\eta_p^l(z_i'^l) - \eta_p^l(\hat{z}_i'^l))^2] + \tag{8}$$

$$\frac{1}{M} \sum_{i=1}^M [(\eta_g^u(z_i^u) - \eta_g^u(\hat{z}_i^u))^2 + (\eta_p^u(z_i'^u) - \eta_p^u(\hat{z}_i'^u))^2], \tag{9}$$

where $\hat{z}_i$ is the feature embedding of the transformed image $\hat{x}_i$. Without enforcing the consistency, we may have different ranking statistics for $\hat{z}_i$ and $z_i$, which will lead to diffferent $s_{ij}$ for the same images under different augmentation, confusing the training.

In summary, the overall loss function used to train our model is

$$\mathcal{L} = \mathcal{L}_{\text{BCE}} + \mathcal{L}_{\text{sKLD}} + \mathcal{L}_{\text{CE}} + \omega(t)\mathcal{L}_{\text{MSE}}, \tag{10}$$

where $\omega(t) = \lambda e^{-5(1-\frac{t}{r})^2}$ is a ramp-up function as widely used in the literature [46, 33, 18] with $\lambda \in \mathbb{R}_+$. $t$ and $r$ are the current time step and the ramp-up length respectively.

After training, as the two branches already agree with each other, we simply take the output of the global branch as the final prediction. In particular, for each unlabelled image, we take the index of the max value in the softmax output as its cluster assignment. It is also worth noting that the memory banks and ranking statistics are no longer needed at inference time.

# 4 Experimental results

## 4.1 Experimental setup

**Benchmark and evaluation metric.** We follow [18] to validate our method on a variety of benchmark datasets for generic image classification including CIFAR-10 [31], CIFAR-100 [31] and ImageNet-1K [11]. For ImageNet-1K, three 30-class unlabelled splits are used in the experiments and the average performance is reported. We further experiment with ImageNet-100 [47] which has less number of labelled classes but the same number of unlabelled classes as ImageNet-1K. In addition, we also validate our method on the more challenging fine-grained datasets including CUB-200 [49], Stanford-Cars [30], and FGVC aircraft [36]. One key assumption for novel category discovery is that the labelled classes are relevant to the unlabelled ones.

Due to the diversity of classes in generic image recognition datasets, it is not easy to tell the relevance between classes, though it is implicitly contained in the rules used to identify classes during data curation. In contrast, the relevance of the fine-grained datasets is explicitly determined by the fact that all classes in a fine-grained dataset belong to the same *entry level* class, *e.g.*, birds, cars, and airplanes for the above fine-grained datasets. Besides, the fine-grained datasets pose more challenges for novel category discovery because of the high inter-class similarity. In this case, spatial local details become crucial clues to distinguish them. The labelled and unlabelled splits are summarized in table 1.

Table 1: **Data splits in the experiments.**

|  | labelled | unlabelled |
|---|---|---|
| CIFAR-10 | 5 | 5 |
| CIFAR-100 | 80 | 20 |
| ImageNet-1K | 882 | {30, 30, 30} |
| ImageNet-100 | 70 | 30 |
| CUB-200 | 160 | 40 |
| Stanford-Cars | 156 | 40 |
| FGVC-aircraft | 81 | 21 |

We follow the standard practice in the literature to adopt clustering accuracy (ACC) on the unlabelled data as the metric for evaluation. It is defined as $\frac{1}{N_t}\sum_{i=1}^{N_t}\mathbb{1}(y_i^* = h(y_i))$ where $h$ is the optimal permutation obtained by Hungarian algorithm [32] that can match the clustering assignment $y_i$ with the ground-truth label $y_i^*$, and $N_t$ is the number of unlabelled images.

**Implementation details.** We follow [18] to initialize our model with parameters pretrained by self-supervised learning. Our default model is realized based on ResNet50 [23]. Namely, the first three macro blocks of ResNet50 are the feature extractor, $f_\theta$, which is initialized with MoCov2 [8] pretrained on ImageNet via self-supervision and are frozen during training for novel category discovery. The last macro block is duplicated into two as the projection layers $\psi_p$ and $\psi_g$, for local and global branches respectively. Each of them is followed by two linear heads, $\eta^l$ and $\eta^u$, for labelled and unlabelled data respectively. For a fair comparison with prior methods, we also experiment using ResNet18 with RotNet [17] initialization. For all our experiments, we use SGD with momentum [44] as the optimizer and use a batch size of 128 for labelled data and 64 for unlabelled data. The temperature parameter $\tau$ is set to 0.07, and the size of the three memory-banks are all set to 2048 for memory efficiency and also considering the relatively limited number of images in the fine-grained datasets. For the dual ranking statistics, we set $k = 5$ for the global branch following [18], and $k = 30$ for the local branch to include more local parts into consideration. Our experiments are performed using GTX 1080Ti GPUs.

## 4.2 Comparison to the state-of-the-art

**Comparison on generic image classification datasets.** In table 2, we compare our method with baselines and state-of-the-art methods for novel category discovery on generic image classification datasets CIFAR-10, CIFAR-100, and ImageNet following the same protocol as RankStat [18] for

fairness. Our method achieves state-of-the-art results on all datasets. It can be seen that our method significantly outperforms $k$-means, KCL [25], MCL [26], and DTC [20]. Notably, our method outperforms the previous state-of-the-art [18] on the most challenging ImageNet-1K dataset by a significant margin of $6.4\%$. The improvements on CIFAR-10 and CIFAR-100 are smaller. We hypothesize that this is due to the spatial resolution discrepancy. The image resolution of CIFAR-10 and CIFAR-100 is only $32 \times 32$, which leads to very small spatial local features. Thus, the local branch cannot offer much more useful information than the global branch. In contrast, the input resolution of ImageNet-1K is $224 \times 224$. Hence, a larger spatial feature map can be obtained to take more advantages of the local branch.

Table 2: **Comparison of novel category discovery on generic classification datasets.** For fair comparison, our method uses ResNet18 [23] backbone initialized with RotNet [17] following [18].

| No | Method | CIFAR-10 | CIFAR-100 | ImageNet-1K |
|----|--------|----------|-----------|-------------|
| (1) | $k$-means [35] | 72.5±0.0% | 56.3±1.7% | 71.9% |
| (2) | KCL [25] | 66.5±3.9% | 14.3±1.3% | 73.8% |
| (3) | MCL [26] | 64.2±0.1% | 21.3±3.4% | 74.4% |
| (4) | DTC [20] | 87.5±0.3% | 56.7±1.2% | 78.3% |
| (5) | RankStat [18] | 90.4±0.5% | 73.2±2.1% | 82.5% |
| (6) | Ours | **91.6±0.6%** | **75.3±2.3%** | **88.9%** |

**Comparison on fine-grained image classification datasets.** In table 3, we compare with other methods on fine-grained image classification datasets. In these datasets, the difference of holistic structure between classes is small, and the classes are mostly differentiated by the local details. Thus, only looking at the global feature descriptors is far from enough for the fine-grained scenario, where the local information plays a vital role. Comparing rows 2–3 with row 5, it can be observed that our proposed method substantially outperforms previous state-of-the-art models that only use global features. For example, our method shows $8.3\%$, $8.1\%$ and $4.1\%$ improvements respectively on CUB-200, Stanford-Cars and FGVC-Aircraft over RankStat [18]. To further verify the effectiveness of our local branch, we carry out another experiment by dropping the global branch (row 4 in table 3). Dropping the global branch will disable the mutual knowledge distillation accordingly. It can be seen that using the local branch alone already yields a notable improvement over RankStat [18] (row 3 vs row 4). Our full method establishes the new state-of-the-art.

Table 3: **Comparison of novel category discovery on fine-grained classification datasets.** "Ours w/o global" means our proposed method without global branch and mutual distillation.

| No | Method | CUB-200 | Stanford-Cars | FGVC-Aircraft |
|----|--------|---------|---------------|---------------|
| (1) | $k$-means [35] | 20.4 ± 1.1% | 31.4 ± 0.9% | 44.7 ± 1.3% |
| (2) | DTC [20] | 33.6 ± 0.7% | 46.5 ± 2.4% | 58.7 ± 1.2% |
| (3) | RankStat [18] | 39.5 ± 1.7% | 53.8 ± 2.0% | 66.3 ± 0.7% |
| (4) | Ours w/o global | 43.1 ± 2.3% | 56.8 ± 2.3% | 67.3 ± 1.0% |
| (5) | Ours full | **47.8 ± 2.4%** | **61.9 ± 2.5%** | **70.4 ± 0.9%** |

## 4.3 Ablation study

**Effectiveness of different components.** In table 4, we ablate different components of our method. It is clear that all components in our method are effective as removing any of them will cause the performance drop. Without BCE loss, our dual ranking statistics is disabled and no pair-wise pseudo labels are transferred from the labelled data. Therefore, the parameters within $\eta^u$ remain untrained, leading to poor performance equivalent to a random baseline. The sKLD loss also has a strong impact on the final performance. Without it, the performance drops $8.0$–$11.3\%$ absolute ACC. This performance drop shows that the information exchange between the global and the local branches are crucial. The consistency loss also plays an important role, as the performance without consistency drops $9.9$–$12.2\%$ absolute ACC. Similar to [18, 20] the cross-entropy loss, consistency loss and self-supervision are also important for our method. Having all these components in a unified framework, our full method obtains the best performance.

Table 4: **Effectiveness of different components of our method.** "MSE" means MSE consistency loss; "CE" means cross-entropy loss for training on labelled data; "BCE" means binary cross-entropy loss for training both global and local branches on unlabeled data; "sKLD" means the sKLD loss for mutual distillation between the two branches; "Self-sup." means self-supervised pre-training.

|  | CUB-200 | Stanford-Cars | FGVC-Aircraft | ImageNet-100 |
|---|---|---|---|---|
| Ours w/o BCE | $2.2 \pm 1.3\%$ | $3.1 \pm 0.4\%$ | $5.1 \pm 0.4\%$ | $3.0 \pm 0.3\%$ |
| Ours w/o sKLD | $39.8 \pm 1.8\%$ | $50.6 \pm 2.1\%$ | $60.8 \pm 1.5\%$ | $58.2 \pm 1.2\%$ |
| Ours w/o CE | $41.2 \pm 2.4\%$ | $52.4 \pm 4.3\%$ | $60.2 \pm 2.7\%$ | $59.1 \pm 2.7\%$ |
| Ours w/o MSE | $37.9 \pm 4.5\%$ | $50.6 \pm 6.2\%$ | $58.9 \pm 5.7\%$ | $57.2 \pm 3.6\%$ |
| Ours w/o Self-sup. | $44.3 \pm 3.5\%$ | $58.2 \pm 1.8\%$ | $67.4 \pm 1.3\%$ | $65.3 \pm 1.3\%$ |
| Ours full | $\mathbf{47.8 \pm 2.4\%}$ | $\mathbf{61.9 \pm 2.5\%}$ | $\mathbf{70.4 \pm 0.9\%}$ | $\mathbf{69.4 \pm 2.1\%}$ |

**Different configurations of the two-branch design.** Our two-branch design allows mutual knowledge distillation between local and global branches. In table 5 we compare different branch configurations. Row 1 and row 2 represent the global and local single-branch baselines. As can be seen, the local branch configuration performs better than the global one which is the configuration of [18], demonstrating the local ranking statistics with the part dictionary is more effective than that with global features. By introducing another branch of the same type (row 1 → row 3, row 2 → row 4) to incorporate mutual knowledge distillation, the performance can be consistently boosted, which further verifies the effectiveness of our mutual knowledge distillation method for novel category discovery. Mutual knowledge distillation between two local branches is more effective than the counterpart between two global branches. Row 5 presents the results of mutual distillation between global branch and local branch, showing best performance, which corroborates that mutual knowledge distillation between local and global branches enables them to complement each other.

Table 5: **Different configurations of the two-branch design.**

| No | Configuration | | CUB200 | Stanford-Cars | ImageNet-100 |
|---|---|---|---|---|---|
| (1) | global | - | $39.5 \pm 1.7\%$ | $53.8 \pm 2.0\%$ | $62.5 \pm 1.2\%$ |
| (2) | local | - | $43.1 \pm 0.9\%$ | $56.8 \pm 1.7\%$ | $64.2 \pm 1.6\%$ |
| (3) | global | global | $41.2 \pm 0.8\%$ | $54.6 \pm 0.7\%$ | $63.2 \pm 0.9\%$ |
| (4) | local | local | $44.7 \pm 1.1\%$ | $57.9 \pm 0.5\%$ | $65.7 \pm 1.4\%$ |
| (5) | global | local | $\mathbf{47.8 \pm 2.4\%}$ | $\mathbf{61.9 \pm 2.5\%}$ | $\mathbf{69.4 \pm 2.1\%}$ |

**Effects of larger memory banks.** It has been shown in [21] that increasing the size of the memory bank can improve self-supervised representation learning. In our experiments, we set the memory sizes of $\mathcal{B}_p$, $\mathcal{B}_g$ and $\mathcal{V}$ to 2048 for training efficiency. In table 6, we show the results by increasing the memory sizes. As can be seen, similar to [21], the memory size and the performance is positively correlated. With more instances or parts in the memory banks, more useful information can be used for mutual knowledge distillation ($\mathcal{B}_p$ and $\mathcal{B}_g$) and local part ranking statistics measure ($\mathcal{V}$), thus improving the results. However, with a relatively small memory size of 2048, our method already achieves promising results. Note that due to the limited number of images in CUB-200 and Stanford-Cars, the performance boost with the increase of bank size for $\mathcal{B}_p$ and $\mathcal{B}_g$ quickly reaches a plateau. While for CIFAR-10 and ImageNet-100, more instances can be enqueued to foster the knowledge distillation. In terms of $\mathcal{V}$, the performance is consistently improved with the increase of the size of $\mathcal{V}$ for all datasets. Due to the larger image resolution in CUB-200 and Stanford-Cars, more useful part features can be extracted from each image for local ranking statistics. Therefore, the performance is less affected by the smaller number of images. This further demonstrates the effectiveness of our local part-level ranking statistics.

**Different sampling methods for memory bank $\mathcal{V}$.** The part memory bank $\mathcal{V}$ stores the part-level features for the local-comparison branch, and is updated in a FIFO manner. Here we compare different ways of maintaining $\mathcal{V}$. Our default choice is to randomly sample one part-level feature from each image in a training mini-batch. However, random sampling may not select the most informative part-level features from the image, so an alternative is to select part-level features based on class-activation-map (CAM) [59]. Instead of random sampling, the part with the highest response is selected by CAM. Rather than sampling only a single part from each image, we can simply use all

Table 6: **Increasing the size of memory bank.** "SCars" denotes Stanford-Cars dataset; "IM-100" denotes ImageNet-100 dataset.

(a) ACC with varying size for $\mathcal{B}_p$ and $\mathcal{B}_g$.

| Bank size | 2048 | 4096 | 8192 | 16384 | 32768 |
|---|---|---|---|---|---|
| CIFAR-10 | 91.6% | 92.3% | 92.4% | 92.7% | 93.0% |
| CUB-200 | 47.8% | 48.3% | 48.4% | 48.5% | 48.4% |
| SCars | 61.9% | 62.4% | 62.8% | 62.9% | 62.9% |
| IM-100 | 69.4% | 70.2% | 70.8% | 71.1% | 71.3% |

(b) ACC with varying size for $\mathcal{V}$.

| Bank size | 2048 | 4096 | 8192 | 16384 | 32768 |
|---|---|---|---|---|---|
| CIFAR-10 | 91.6% | 91.7% | 91.7% | 91.9% | 92.0% |
| CUB-200 | 47.8% | 48.5% | 48.7% | 49.2% | 49.3% |
| SCars | 61.9% | 62.7% | 63.4% | 63.6% | 63.7% |
| IM-100 | 69.4% | 70.5% | 71.2% | 71.3% | 71.4% |

the parts for each image. We compare these three different choices in table 7. It can be seen that using all the parts causes slight performance drop. This is reasonable because using all parts introduces redundancy in $\mathcal{V}$, making the local comparison more prone to noise. On the other hand, selecting the most activated part using CAM shows a better performance than the random selection, indicating that more informative parts in the memory bank can enable better local comparison. As the gap is not significant, we use the random sampling as our default choice for its simplicity and efficacy.

Table 7: **Effects of using different way to update $\mathcal{V}$.** "Random" means random part selection; "All" means using all parts; "CAM" means selecting the most activated parts using CAM [59].

|  | CIFAR-100 | ImageNet-100 | CUB-200 | Stanford-Cars |
|---|---|---|---|---|
| Random | $75.3 \pm 2.3\%$ | $69.4 \pm 2.1\%$ | $47.8 \pm 2.4\%$ | $61.9 \pm 2.5\%$ |
| All | $76.0 \pm 2.6\%$ | $68.7 \pm 2.3\%$ | $46.5 \pm 1.9\%$ | $61.0 \pm 2.4\%$ |
| CAM | $76.5 \pm 2.4\%$ | $70.3 \pm 2.4\%$ | $48.5 \pm 1.8\%$ | $62.4 \pm 2.1\%$ |

**Unknown number of novel classes.** In a more realistic setting, the unlabelled class number $C^u$ may not be known. In this case, we can apply existing methods to estimate category number in the unlabelled data, such as the algorithm from DTC [20], before adopting our method. In table 8 we provide the results of novel category discovery using the estimated number of classes on CUB-200/ImageNet-100 by DTC [20]. We denote the estimated class number as $\hat{C}^u$. As shown in table 8, with the estimated class number by DTC, our model also obtains a significantly better performance than RankStat [18].

Table 8: **Results with estimated class numbers and ground truth class numbers.**

|  | CUB-200 | | ImageNet-100 | |
|---|---|---|---|---|
| Class numbers | $C^u = 40$ | $\hat{C}^u = 43$ | $C^u = 30$ | $\hat{C}^u = 32$ |
| RankStat [18] | 39.5±1.7% | 38.2±2.1% | 66.3±0.7% | 65.0±1.2% |
| Ours | **47.8±2.4%** | **44.6±2.6%** | **70.4±0.9%** | **68.8±1.5%** |

Please refer to the supplementary for more results and analysis. Our code can be found at `https://github.com/DTennant/dual-rank-ncd`.

## 5   Conclusion

We have introduced a two-branch learning framework for novel category discovery, with one branch focusing on local part-level information and the other branch focusing on overall characteristics. To transfer knowledge from labelled data to unlabelled data, we proposed to use local ranking statistics by maintaining a local part dictionary build on-the-fly for training, together with the global ranking statistics. We further introduced a mutual knowledge distillation method for information exchange and encouraging the agreement between the two branches. We thoroughly evaluated our method on generic image classification datasets as well as fine-grained recognition datasets, achieving state-of-the-art performance.

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
