# Novel Visual Category Discovery with Dual Ranking Statistics and Mutual Knowledge Distillation –Supplementary Material–

Bingchen Zhao[1]    Kai Han[2,3,4*]

[1]Tongji University  [2]The University of Hong Kong  [3]Google Research  [4]University of Bristol
zhaobc.gm@gmail.com  kaihanx@hku.hk

## Contents

---

* Corresponding author.

## A Implementation details

We use the MoCo v2 [2] self-supervised model pretrained with 800 epochs on ImageNet-1K dataset to initialize our model for experiments on all datasets, except CIFAR10/100 and ImageNet-1K, for which we use the RotNet pretrained model provided by [4] for fair comparison. We train our model for 200 epochs on CIFAR10/100, CUB-200, Stanford-Cars, FGVC-Aircraft, and ImageNet-100 datasets, and 90 epochs on ImageNet-1K dataset. The initial learning rate is set to 0.1 for all datasets except ImageNet-1K, and is scheduled to decay by a factor of 10 at the 170th epochs. For ImageNet-1k dataset, the initial learning rate is set to 0.03, and is scheduled to decay by a factor of 10 at the 30th and 60th epochs following the common practice. For the consistency regularization term, we set the hyper-parameters in the ramp-up function following [4] on CIFAR-10/100 and ImageNet-1k. For the experiments on CUB-200, Stanford-Cars, FGVC-Aircraft, and ImageNet-100, we set $\lambda = 50$ and $r = 150$.

## B Unlabelled data containing both seen and unseen classes

Here we provide the results under the open world semi-supervised setting [1], where the unlabelled data does not only contain the novel classes, but also the seen classes in the labelled set, which can be regarded as an extended setting of novel category discovery. In this case, the model needs to recognize seen classes and discover novel classes simultaneously in the unlabelled data. To enable our model to handle this scenario, we extend the classification head $\eta^u$ to have the output with a dimension of $C^l + C^u$, with the first $C^l$-dimension corresponding to the labelled seen classes and the last $C^u$-dimension corresponding to the unlabelled new classes. $\eta^u$ is trained with cross-entropy loss on the labelled data and binary cross-entropy loss on the unlabelled data. To prevent the classifier from being biased towards the known classes, we follow the very recent preprint [1] to include two regularization techniques to train our model. The first one is to perform the $\ell_2$ normalization on both the classifier weights and the features, and the second one is to optimize the KL divergence between the predicted logits outputed by $\eta^u$ and a uniform distribution. In table 1, we compare our method with DTC [5], RankStat [4], and ORCA [1] on the ImageNet-100 dataset. Our method significantly outperforms all others on novel classes, and performs on par with ORCA [1] on the seen classes.

Table 1: **Comparison under open world semi supervise setting.**

| No | Split (seen/novel) | 50/50 | | 25/75 | |
|---|---|---|---|---|---|
| | Classes | Seen | Novel | Seen | Novel |
| (1) | DTC [5] | 25.6% | 20.8% | 23.5% | 18.1% |
| (2) | RankStat [4] | 63.7% | 47.9% | 52.4% | 43.8% |
| (3) | ORCA [1] | **89.1%** | 72.1% | 89.4% | 67.4% |
| (4) | Ours | 87.6% | **76.0%** | **89.5%** | **71.3%** |

## C Varying $k$ for ranking statistics

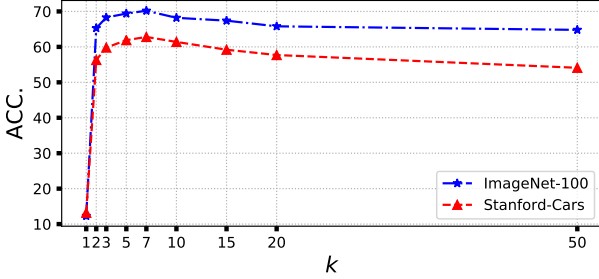

Figure 1: Performance with varying $k$ for global ranking statistics.

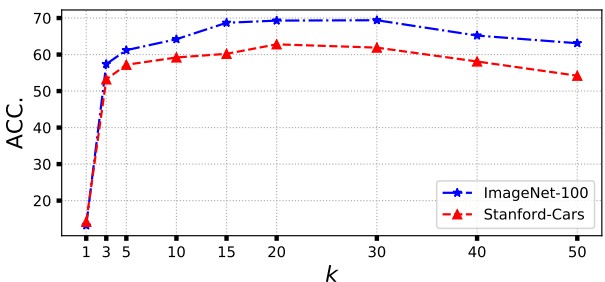

Figure 2: Performance with varying $k$ for local ranking statistics.

In fig. 1 and fig. 2, we report the results with varying $k$ for the soft ranking statistics on ImageNet-100 and Stanford-Cars datasets. Our default choice is 5 for the global one following [4] and 30 for the local one to take more local parts into consideration. It can be seen, except the extreme case with very small $k$ (*e.g.* $k = 1$), the results are generally stable, further corroborating the robustness of ranking statistics.

## D   A single-branch variant of our method with dual ranking statistics

Given that local ranking statistics is more strict than global ranking statistics in pairwise verification, the positives obtained by local ranking statistics is more reliable while the negatives obtained by global ranking statistics is more reliable. This motivates us to study the single-branch variant of our method by simply providing positive and negative pairs using local and global ranking statistics respectively, in which our mutual learning is no longer applicable. We report the results in the last row of table 2. We also validate other possible ways of providing positive and negative pairs. It can be seen, using local and global ranking statistics to provide positives and negatives respectively yields better performance than all other combinations. However, the performance of this single-branch variant still notably lags behind our two-branch model (see table 4 in the main paper), due to the absence of mutual learning.

Table 2: **Comparison of using different branches for positive and negative.**

| Positive source | Negative source | CUB-200 | ImageNet-100 |
|---|---|---|---|
| global | global | 39.5% | 62.5% |
| local | local | 43.1% | 64.2% |
| global | local | 38.6% | 61.9% |
| local | global | **44.3%** | **64.7%** |

## E   Qualitative results

In fig. 3, we visualize the t-SNE [10] projection of the global feature embeddings on the unlabelled data. It is clear that along with the training the features on unlabelled data become more and more discriminative for both ImageNet-100 and Stanford-Cars. We can also observe that MoCo v2 [2] initialization appears to be better for ImageNet-100 dataset than Stanford-Cars. This is reasonable because the pretraining of MoCo v2 is conducted on ImageNet-1K, which contains ImageNet-100 as a subset. On Stanford-Cars dataset, though the MoCo v2 initialization does not apppear to be good, our method can still learn discriminative features after training.

## F   Comparing ranking statistics with cosine similarity

Here, we present results using ranking statistics and cosine similarity for pseudo label generation in our two-branch framework. For ranking statistics, we experiment on both the hard version introduced

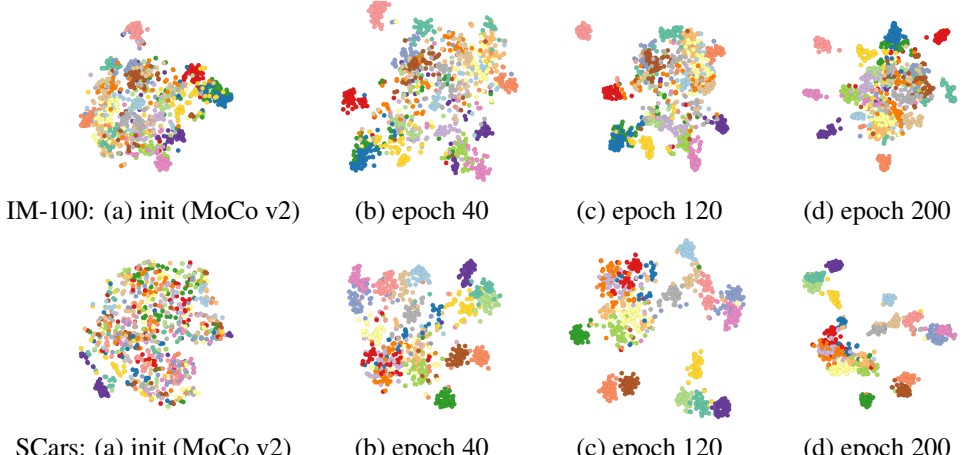

Figure 3: **Evolution of the t-SNE during the training.** Colors of data points denote their ground-truth labels. The first row shows the t-SNE visualization for ImageNet-100 dataset, and the second row shows that for Stanford Cars dataset.

Table 3: **Comparing ranking statistics (RS) with cosine similarity (CS).**

| Global | Local | Mode | CUB-200 | Stanford-Cars | ImageNet-100 |
|--------|-------|------|---------|---------------|--------------|
| CS | - | hard | $38.5 \pm 0.6\%$ | $52.1 \pm 1.1\%$ | $61.8 \pm 1.3\%$ |
| - | CS | hard | $42.8 \pm 1.0\%$ | $52.5 \pm 0.8\%$ | $61.5 \pm 1.0\%$ |
| CS | CS | hard | $44.8 \pm 1.3\%$ | $59.1 \pm 2.0\%$ | $67.4 \pm 0.8\%$ |
| CS | - | soft | $39.1 \pm 0.5\%$ | $52.3 \pm 1.0\%$ | $61.7 \pm 1.1\%$ |
| - | CS | soft | $42.9 \pm 0.8\%$ | $53.2 \pm 0.7\%$ | $62.2 \pm 1.3\%$ |
| CS | CS | soft | $43.7 \pm 2.2\%$ | $59.4 \pm 1.2\%$ | $67.5 \pm 1.1\%$ |
| RS | - | hard | $39.4 \pm 1.2\%$ | $53.2 \pm 1.7\%$ | $62.3 \pm 1.4\%$ |
| - | RS | hard | $43.3 \pm 0.3\%$ | $56.4 \pm 1.3\%$ | $63.7 \pm 1.5\%$ |
| RS | RS | hard | $47.1 \pm 1.3\%$ | $60.8 \pm 1.7\%$ | $68.9 \pm 1.2\%$ |
| RS | - | soft | $39.5 \pm 1.7\%$ | $53.8 \pm 2.0\%$ | $62.5 \pm 1.2\%$ |
| - | RS | soft | $43.1 \pm 0.9\%$ | $56.8 \pm 1.7\%$ | $64.2 \pm 1.6\%$ |
| RS | RS | soft | $\mathbf{47.8 \pm 2.4\%}$ | $\mathbf{61.9 \pm 2.5\%}$ | $\mathbf{69.4 \pm 2.1\%}$ |

in [4] and the soft version we use in this paper (see Sec. 3.1 in the main paper). We also carry out experiments using "hard" and "soft" cosine similarity. For the "hard" cosine similarity, we simply adopt a threshold (0.9 in our experiments) on the score to get binary pseudo labels. While for the "soft" cosine similarity, we directly take the score as soft pseudo labels. The results are presented in table 3. For both cosine and ranking statistics, the hard mode and soft mode perform comparably well, with soft ranking statistics showing slightly better performance than hard ranking statistics. While in all cases, ranking statistics performs better than cosine similarity, demonstrating the robustness of ranking statistics. We choose to use soft ranking statistics because we believe the continuous similarity better reflect the actually similarity of objects than the binary score. This is important for the pairs with a similarity score around 0.5, for which the binary score is not very reliable.

# G Effects of an additional supervised finetuning stage

By default, the training of our model consists of two stages. The first stage is self-supervised pretraining, and the second stage is joint training for novel category discovery on labelled and unlabelled data. RankStat [4] also adopts an additional supervised finetuning stage before joint training. We validate the effectiveness of such an additional supervised finetuning stage for our

model. Specifically, after the self-supervised pretraining stage, we freeze the first three macro block of ResNet [6], and finetune the last macro block with cross-entropy loss on the labelled categories, before the joint training. Results are presented in table 4. It can be seen that this additional finetuning stage does not bring obvious gains for our model, further demonstrating the effectiveness of our design.

Table 4: **Comparison of adding the additional supervised finetuning stage.**

| Method | CIFAR-10 | CIFAR-100 | CUB-200 | Stanford-Cars | ImageNet-1K |
|---|---|---|---|---|---|
| RankStat [4] | 90.4±0.5% | 73.2±2.1% | 39.5 ± 1.7% | 53.8 ± 2.0% | 82.5% |
| Ours w/o sup. | **91.6±0.6%** | 75.3±2.3% | 47.8 ± 2.4% | **61.9 ± 2.5%** | **88.9%** |
| Ours w/ sup. | 90.8±0.7% | **75.9±2.2%** | **48.2 ± 2.1%** | 61.7 ± 2.1% | 88.4% |

## H  Loss for mutual knowledge distillation

In section 3.2, we adopt the symmetric Kullback-Leibler Divergence (sKLD) as the loss for mutual knowledge distillation between the two branches. Another widely applied loss for mutual learning is the Jensen-Shannon Divergence(JSD) loss. In table 5 we compare our model by training with these two different losses. Both losses are equally valid for our model.

Table 5: **Different losses for mutual learning.**

| Method | CIFAR-10 | CIFAR-100 | CUB-200 | Stanford-Cars | ImageNet-1K |
|---|---|---|---|---|---|
| Ours w/ sKLD | 91.6±0.6% | 75.3±2.3% | 47.8 ± 2.4% | 61.9 ± 2.5% | 88.9% |
| Ours w/ JSD | 91.4±0.4% | 75.6±2.5% | 47.9 ± 2.6% | 61.7 ± 2.4% | 88.8% |

## I  Limitations and potential negative societal impacts

Although our method can achieve state-of-the-art performance on public datasets, the performance still notably lags behind fully supervised models. Moreover, real-world data is much more complicated than the curated data used in our experiments. Therefore, under safety-critical situations, such as autonomous driving and medical image analysis, our method is not expected to provide reliable enough inference, especially when the unlabelled data is largely different from the labelled data or contains unpredictable noises. Hence, careful validation on the specific application scenario should be carried out, before the deployment on any real-world environment.

## J  License of used datasets

All the datasets used in this paper are permitted for research use. CIFAR-10 and CIFAR-100 datasets [8] are released under the MIT license, allowing use for research purposes. The terms of access of the ImageNet dataset [3] allow the use for non-commercial research and educational purposes. Similar to ImageNet, the Stanford Cars [7] allows the use for research purposes. The FGVC aircraft [9] dataset was made available exclusively for non-commercial research purposes by the authors. The CUB-200 [11] dataset also allows research use.