# OpenReview forum: "Novel Visual Category Discovery with Dual Ranking Statistics and Mutual Knowledge Distillation"
_NeurIPS.cc/2021/Conference — NeurIPS 2021 Poster_

### Official Review · Reviewer_72SS · 2021-07-13

**Rating:** 7
**Confidence:** 4

**Summary:**

The paper proposes a new technique for visual class discovery, which relies on soft label assignment via ranking statistics, computed from both global features and local/part patterns. One of the most novel components is a JSD loss to align global/part features; other components include ranking statistics to compute soft pseudo labels, and MSE for consistency of predictions under different transformations.

**Limitations And Societal Impact:**

Not discussed, not necessary

**Main Review:**

The approach seems sound. Some components are more novel (ranking statistics for labels, global/local alignment) and others are less novel. The contribution of each part seems properly verified through ablations. Results overall are strong enough, although margins to the baselines are not huge. In particular, both ranking statistics and global/local alignment are shown to have an impact. The particular manner of doing distillation, local-to-global, is also ablated. Multiple evaluation settings are shown, i.e. standard (coarse) classification as well as fine-grained.

Minor questions:
- the s_{ij} notation is a bit strange--usually labels are for one sample?
- what is z_i^{'u} i.e. what is the ' here? (above Eq. 4)
- there shouldn't be a sentence break on L134

**Time Spent Reviewing:**

1

---

> ### Author Response · Authors · 2021-08-09
> **Answer to Reviewer 72SS**
>
> > Q1: the s_{ij} notation is a bit strange--usually labels are for one sample?
> >
> > A1: In our method, the pseudo-labels are generated for each pair of samples, indicating whether the two samples are from the same novel class or not. We follow the notation used in previous works like [A,B,C] for the novel category discovery task.
>
> > Q2: what is z_i^{'u} i.e. what is the ' here? (above Eq. 4)
> >
> > A2: It denotes the feature extracted from the local part branch by ${\bar \psi}_p$. We will further clarify this in the final version.
>
> > Q3: there shouldn't be a sentence break on L134
> >
> > A3: Thanks. We will fix it in the final version.
>
> [A] Learning to Discover Novel Visual Categories via Deep Transfer Clustering, ICCV 2019
>
> [B] Automatically Discovering and Learning New Visual Categories with Ranking Statistics, ICLR 2020
>
> [C] AutoNovel: Automatically Discovering and Learning Novel Visual Categories. TPAMI 2021

---

### Official Review · Reviewer_iVtL · 2021-07-16

**Rating:** 8
**Confidence:** 5

**Summary:**

This paper considers the problem of novel class discovery (NCD). Specifically, the authors propose a Dual Ranking Statistics method that can leverage both global and local factors for learning novel classes. In addition, a Mutual Knowledge Distillation approach is proposed to explore the relationship between the global and local branches, which can further improve the results. Experiments on several datasets, especially on fine-grained cases, show the benefit of the proposed method.

**Limitations And Societal Impact:**

Yes

**Main Review:**

Pros:

+ Writing: This paper is well written and easy to follow. The overall paper is well-organized. Figures and Tables are clear to show the methods and comparisons.

+ Novelty: This paper considers an interesting problem, i.e., novel class discovery. In addition, a fine-grained setting is introduced in this paper, which is not studied in previous NCD works. For the method, this paper introduces two simple but effective approaches, which explicitly take the advantage of both the global and local factors and obtain consistent improvement.

+ Experiments: This paper presents extensive experiments to show the effectiveness of the components of the proposed method. The comparisons with other methods are comprehensive and fair. In addition, the authors also provide the experiments on more difficult settings: unknown number of novel classes and open-set unlabeled data.

Cons/Concerns:

I have some questions about the method and the experiment.

- Did you use the proposed losses for the labeled data, especially the local loss? If no, how about the results of using them for the labeled data. Will this improve the discrimination of the representation?

- For the part dictionary memory bank, why not save all the parts of a sample into the memory? Could you provide the difference between using a random part of a sample and using all parts of a sample?

- Saving random/all part features into the memory is a bit wild? Is it possible to first find high-response regions (for example the activations) and only save them into the memory?

- The proposed has two branches, which one is used for the testing stage? With JSD loss, I think these two branches may have similar results. However, when removing JSD loss, which one is used?

- In Table 6, it seems that a larger memory size brings higher results. Why not using the larger memory size for all other experiments? Will it lead to a very large computation cost?

- For the self-pretraining, how about the results of removing the self-pretraining? In addition, why use RotationNet for CIFAR while MOCOV2 for others? How about the results of using RotationNet/MOCO for others/CIFAR?

- Which dataset is evaluated in Table 1 of Supple. Is it CIFAR-100?

- Open questions: (1) In the open-set setting (Table 1 of Supple), did you use any technique to recognize the samples of known and unseen classes? If no, will the model assign all the samples to the known classes and give very few values/probabilities to the novel classes? (2) It would be nice that the authors can provide a visualization for the local comparison process. For example, we save many local features in the memory bank, when computing the similarity between each part of a sample and parts in the memory, we visualize the image rejoins of corresponding highly ranked parts.

Overall, this paper considers an interesting problem and presents a novel and effective approach, which achieves the SOTA results. During rebuttal, please mainly answer the first 7 concerns. The authors can also selectively provide more discussions and visualizations for the open questions.

**Post Rebuttal** After reading the response of the authors, my concerns are well-addressed. I also read the comments of other reviewers and I agree with them that this paper presents a novel approach for novel class discovery. Thus, I decided to keep my original rating and tend to give 'accept' to this paper.



**Time Spent Reviewing:**

8 hours

---

> ### Author Response · Authors · 2021-08-09
> **Answer to Reviewer iVtL**
>
> > Q1: Did you use the proposed losses for the labeled data, especially the local loss? If no, how about the results of using them for the labeled data. Will this improve the discrimination of the representation?
> >
> > A1: We use the same losses (CE, BCE, sKLD) for both branches. For each branch, CE is used on labeled data, BCE is on unlabeled data, and sKLD is on both labelled and unlabelled data, so that we can make the best use of data for training. Our local branch only uses part comparison for pairwise pseudo label generation, while the losses are the same with the global branch and are applied for both labelled and unlabelled data.
>
> > Q2: For the part dictionary memory bank, why not save all the parts of a sample into the memory? Could you provide the difference between using a random part of a sample and using all parts of a sample?
> >
> > A2: Thanks for the suggestion. We have experimented using all parts of a sample into the part dictionary on CUB-200, and the ACC is 46.5, which is slightly lower than using only a random part (i.e. 47.8), we suspect that this is because using all parts will make the part dictionary redundant with many similar parts, and thus the performance of local comparison may be degraded due to less informative parts.
>
> > Q3: Saving random/all part features into the memory is a bit wild? Is it possible to first find high-response regions (for example the activations) and only save them into the memory?
> >
> > A3: Thanks for the helpful suggestion. In the submission, we used the random part for simplicity. Here, we experimented using only the highest response part into the part dictionary. The highest response region is selected using the method proposed in [A].  We obtain an improved ACC of 48.5 on the CUB-200 dataset. We will run experiments on other datasets and report the corresponding results in our final version.
>
> > Q4: The proposed has two branches, which one is used for the testing stage? With JSD loss, I think these two branches may have similar results. However, when removing JSD loss, which one is used?
> >
> > A4: (1) We simply use the global branch for testing, because with JSD loss, the two branches agree with each other after training. (2) For testing, we can drop the local branch and the three memory banks, leaving the global branch as the simplest model for testing. (3) For the experiment without JSD, we also simply use the global branch for testing.
>
> > Q5: In Table 6, it seems that a larger memory size brings higher results. Why not using the larger memory size for all other experiments? Will it lead to a very large computation cost?
> >
> > A5: In Table 6, we show that the performance can be further improved for practitioners who have enough computation and memory resources by using a larger bank size. However, the improvement of a larger memory bank compared to the additional computation and memory cost is not substantial. Hence, we simply use the relatively small number of 2048 for computational efficiency.
>
> > Q6: For the self-pretraining, how about the results of removing the self-pretraining? In addition, why use RotationNet for CIFAR while MOCOV2 for others? How about the results of using RotationNet/MOCO for others/CIFAR?
> >
> > A6: Indeed, we have already reported results without self-supervised pretraining in row 5 of Table 4 in our submission. The numbers are reasonably good, but consistently worse than those with self-supervised pretraining. We use RotNet for CIFAR for a fair comparison with previous methods. The usage of MoCoV2 on other datasets is for a better baseline performance on the new datasets we experimented in our paper. We have experimented using RotNet pretraining on CUB-200 and Stanford-Cars, the cluster acc are 45.2 and 58.5 respectively, which are inferior to using MoCoV2 for our framework.
>
> > Q7: Which dataset is evaluated in Table 1 of Supple. Is it CIFAR-100?
> >
> > A7: The dataset evaluated in Table 1 of supplementary is ImageNet-100, following the setting in [B]. We will clarify this in the final version.
>
> > Q8: Open questions: (1) In the open-set setting (Table 1 of Supple), did you use any technique to recognize the samples of known and unseen classes? If no, will the model assign all the samples to the known classes and give very few values/probabilities to the novel classes? (2) It would be nice that the authors can provide a visualization for the local comparison process. For example, we save many local features in the memory bank, when computing the similarity between each part of a sample and parts in the memory, we visualize the image rejoins of corresponding highly ranked parts.
> >
> > A8: (1) We didn’t use any technique to recognize unseen classes. For this particular setting, to overcome the issue of the model biasing towards the known classes, following [B], the classifier weight and features are l2-normalized during training, and we also use the same regulation term as in [B] to optimize the KL divergence between the predicted logits and a uniform distribution to prevent the known class to dominate the prediction. We will include detailed discussion about this setting and clarify these details in the final version. (2) Thanks for the suggestion. We will include the visualization for the local comparison process in the final version.
>
> [A] Learning deep features for discriminative localization, CVPR 2016
>
> [B] Open-World Semi-Supervised Learning, Arxiv 2021

---

### Official Review · Reviewer_uqsh · 2021-07-16

**Rating:** 6
**Confidence:** 3

**Summary:**

The paper tackles novel class discovery of unlabeled data via transfer learning-based two-brach learning with pairwise pseudo-labels by two different criteria: similarity on global descriptor and local features. The distinct criteria generate more robust pseudo labels than the single ones and alleviate the negative impact from false pseudo-labels in a complementary manner. The proposed mutual knowledge distillation further facilitates effective mutual learning between two branches. Extensive experiments on standard classification datasets and more challenging fine-grained datasets show that their method has significant advantages over SOTA owing to robust pseudo-labels by dual rank statistics.


**Limitations And Societal Impact:**

1. I wonder why the authors do not conduct fine-tuning on labeled data before training with unseen unlabeled data. Does initialization with self-supervised learning generate more robust pseudo-labels than a fine-tuned model in the early training stage? and why? In a similar vein, what initialization is used for the *Ours w/o Self-sup* in Table 4?

2. Some experimental issues.
    * How did the authors conduct validation? Due to the existence of unseen classes in unlabeled data, how to conduct validation is not trivial.
    * Have different distillation approaches been investigated for mutual training? Such as the l2-regularization between $z_i$ and $z^\prime_i$.

3. Equation of Jensen Shannon Divergence (Eq. 6) is wrong. Jensen Shannon Divergence is $\operatorname{JSD}(p, q)=\frac{1}{2} D_{K L}\left(p \| \frac{p+q}{2}\right)+\frac{1}{2} D_{K L}\left(q \| \frac{p+q}{2}\right)$.

**Main Review:**

* The paper is overall well-written and well-organized with clear motivation.

* The setting of the novel class discovery is interesting and should be practically important.

* The experiments are comprehensive and the results show clear improvement over previous SOTA.


**Time Spent Reviewing:**

I read the paper thoroughly, and spent a few days for reviewing.

---

> ### Author Response · Authors · 2021-08-09
> **Answer to Reviewer uqsh**
>
> > Q1: I wonder why the authors do not conduct fine-tuning on labeled data before training with unseen unlabeled data. Does initialization with self-supervised learning generate more robust pseudo-labels than a fine-tuned model in the early training stage? and why? In a similar vein, what initialization is used for the Ours w/o Self-sup in Table 4?
> >
> > A1: Indeed, like [A, B, C], conducting the fine-tuning on labelled data before training with unlabelled data is helpful, however, we found that our method can already produce promising results without this step, while adding this extra fine-tuning stage will require more computation. Thus, we simplify the framework by not including it. We experiment on the CUB-200 dataset by adding this extra fine-tuning stage, and the resulting ACC is 48.2. The number w/o this additional stage is 47.8. We will include numbers on the other datasets with this additional fine-tuning stage in our final version. For the initialization of Ours w/o Self-sup experiments in Table 4, we simply use a random initialization.
>
> > Q2: Some experimental issues. (1) How did the authors conduct validation? Due to the existence of unseen classes in unlabeled data, how to conduct validation is not trivial. (2) Have different distillation approaches been investigated for mutual training? Such as the l2-regularization between zi and zi′.
> >
> > A2: (1) We adopt the same validation process as in [A, B, C] for selecting hyperparameters using only the labelled data. Specifically, we further split the $C^l$ known classes into two subsets, an anchor probe set (keeping the labels) and a validation probe set (dropping the labels, and pretending to be unlabelled). The number of classes in the validation probe set is set to $C^u$. We train our model on these two sets to determine the hyperparameters by examining the performance on the validation probe set. These parameters are then used to train the model with $C^l$ labelled classes and $C^u$ unlabelled classes. (2)Thanks for the suggestion. We follow the suggestion to use l2-regularization between zi and i`, and the ACC on CUB-200 and Stanford-Cars are 44.2 and 57.9 respectively, which lag behind the numbers 47.8 and 61.9 using the symmetric KLD loss. We further explored the JSD loss as pointed out in Q3 below.
>
> > Q3: Equation of Jensen Shannon Divergence (Eq. 6) is wrong.
> >
> > A3: Thanks for pointing out our carelessness here. Eq. 6 is actually the symmetric KL Divergence (sKLD). We will correct this in the final version. Additionally, we also experiment with the proper JSD on CUB-200, and obtain a slightly better ACC of 48.2. Moreover, by further introducing the extra fine-tuning stage on labelled data as mentioned in Q1, we can achieve a further boost of ACC to 48.4 on CUB-200. We appreciate the constructive comment and will carry out the corresponding experiments on other datasets and include the results in our final version.
>
>
> [A] Learning to Discover Novel Visual Categories via Deep Transfer Clustering, ICCV 2019
>
> [B] Automatically Discovering and Learning New Visual Categories with Ranking Statistics, ICLR 2020
>
> [C] AutoNovel: Automatically Discovering and Learning Novel Visual Categories. TPAMI 2021

---

### Official Review · Reviewer_wP3g · 2021-07-21

**Rating:** 7
**Confidence:** 4

**Summary:**

This paper introduces a new method to discover novel visual categories from unlabelled images. The experimental results demonstrate the effectiveness of the proposed method and achieve the state-of-the-art result on fine-grained visual recognition and object classification.

**Limitations And Societal Impact:**

Authors should consider adding more pictures to make the paper easier to read. And add some case studies of the classification results.

**Main Review:**

The paper utilizes dual ranking statistics to transfer knowledge from known classes in the labeled data to the unlabelled data and can generate a more robust pseudo label for learning on the unlabelled data.

Besides, a mutual knowledge distillation method is proposed to allow information exchange and encourage agreement between the local and global branches.

The paper is well organized and easy to follow. The proposed model consists of two branches. Although the usage of the existing binary pseudo label limits the novelty of the model, the dual structure and the distillation to encourage agreement between branches are innovative. The global branch originates from [1] and the local branch includes a FIFO queue as a memory bank to query the ranking statistics as a dynamic object part dictionary.

Typo: a redundant close parenthesis in the rear of Eqn (8).

[1] Han K, Rebuffi S A, Ehrhardt S, et al. Automatically Discovering and Learning New Visual Categories with Ranking Statistics[C]//International Conference on Learning Representations. 2019.

**Time Spent Reviewing:**

2h

---

> ### Author Response · Authors · 2021-08-09
> **Answer to Reviewer wP3g**
>
> > Q1: Adding more pictures to make the paper easier to read
> >
> > A1: Thanks for the suggestion. We will follow the suggestion to add more visualization to better demonstrate the local part comparison process, which is also suggested by Reviewer iVtL, and add more figures to show Qualitative results in the final version.
>
> > Q2: Add some case studies of the classification results
> >
> > A2: We have included a few case studies and ablations in the supplementary. For example, (1) exploring the case where unlabelled data contain both seen and unseen classes, (2) varying the value of k in ranking statistics, (3) creating positive and negative pairs using local and global ranking statistics respectively for a single branch method, and (4)comparison between the pseudo label generated by ranking statistics and cosine similarity. We will add more case studies such as using soft ranking statistics and cosine similarity for both local and global branches in the final version.
>
> > Q3: Typo
> >
> > A3: Thanks. We will fix it.

---

### Author Response · Authors · 2021-08-09
**General response**

We appreciate all reviewers for the unanimously positive comments and constructive suggestions. Individual concerns have been addressed carefully in the response to each reviewer. In the final version, we will revise the paper following the suggestions.

---

### Decision · Program_Chairs · 2021-09-27

**Decision:**

Accept (Poster)

**Comment:**

This paper presents a novel class discovery technique based on dual ranking statistics and mutual knowledge distillation. The main idea is reasonable and shows superior performance compared to existing methods. All reviewers are positive about this paper and there are no particularly negative comments. However, the proposed method is lacking in theoretical support and the experiments are limited to small datasets and/or relatively a small number of unlabeled classes. Overall, I recommend accepting this paper for a poster presentation.